# Race and resource allocation: an online survey of US and UK adults' attitudes toward COVID-19 ventilator and vaccine distribution

Andreas Kappes,[1] Hazem Zohny [iD],[2] Julian Savulescu,[2,3] Ilina Singh,[4] Walter Sinnott-Armstrong,[5] Dominic Wilkinson [iD] [2,6]

¹Psychology, City University of London, London, UK
²Oxford Uehiro Centre for Practical Ethics, Univeristy of Oxford, Oxford, UK
³Centre for Biomedical Ethics, National University of Singapore, Singapore
⁴Psychiatry, University of Oxford, Oxford, UK
⁵Philosophy, Duke University, Durham, North Carolina, USA
⁶Newborn Care Unit, John Radcliffe Hospital, Oxford, UK

**Correspondence to**
Dr Hazem Zohny;
Hazem.zohny@philosophy.ox.ac.uk

## ABSTRACT

**Objective** This study aimed to assess US/UK adults' attitudes towards COVID-19 ventilator and vaccine allocation.

**Design** Online survey including US and UK adults, sampled to be representative for sex, age, race, household income and employment. A total of 2580 participated (women=1289, age range=18 to 85 years, Black American=114, BAME=138).

**Interventions** Participants were asked to allocate ventilators or vaccines in scenarios involving individuals or groups with different medical risk and additional risk factors.

**Results** Participant race did not impact vaccine or ventilator allocation decisions in the USA, but did impact ventilator allocation attitudes in the UK ($F(4,602)=6.95$, $p<0.001$). When a racial minority or white patient had identical chances of survival, 14.8% allocated a ventilator to the minority patient (UK BAME participants: 24.4%) and 68.9% chose to toss a coin. When the racial minority patient had a 10% lower chance of survival, 12.4% participants allocated them the ventilator (UK BAME participants: 22.1%). For patients with identical risk of severe COVID-19, 43.6% allocated a vaccine to a minority patient, 7.2% chose a white patient and 49.2% chose a coin toss. When the racial minority patient had a 10% lower risk of severe COVID-19, 23.7% participants allocated the vaccine to the minority patient. Similar results were seen for obesity or male sex as additional risk factors. In both countries, responses on the Modern Racism Scale were strongly associated with attitudes toward race-based ventilator and vaccine allocations ($p<0.0001$).

**Conclusions** Although living in countries with high racial inequality during a pandemic, most US and UK adults in our survey allocated ventilators and vaccines preferentially to those with the highest chance of survival or highest chance of severe illness. Race of recipient led to vaccine prioritisation in cases where risk of illness was similar.

### STRENGTHS AND LIMITATIONS OF THIS STUDY

⇒ Our scenarios enable comparison of approaches to prioritisation of ventilators and vaccines in the face of scarcity and high demand.

⇒ The survey provides insights into the relative weights given to equity and race compared with medical factors in two different countries.

⇒ To control variables, hypothetical scenarios were not realistic and may not align with lived experience.

⇒ Survey responses permit the quantification of attitudes to prioritisation, but not the reasons behind answers.

⇒ We sought large nationally representative samples, but Hispanic and black participants were underrepresented among US respondents, and members of non-academic minoritised groups were not included in the study design or analysis.

## INTRODUCTION

The coronavirus pandemic has raised contentious ethical questions about the distribution of scarce healthcare resources during a crisis. For example, early concern about the supply of intensive care beds and mechanical ventilators led healthcare systems to develop triage criteria in the event of a shortage.[1 2] Later, states had to decide whom to prioritise for access to the limited supply of vaccines.[3 4] Although different countries have followed different strategies, a prevailing response to these questions has focused on saving the most lives.[3 5–8] For ventilator triage, this entailed giving priority to those with the highest chance of survival.[9] For vaccines, those with the highest risk of severe illness were prioritised.[10]

However, this has been criticised[11] for ignoring evidence that individuals from racial minorities have been disproportionately affected by the pandemic.[12–14] In the USA, black, Hispanic and Indigenous people were more likely to be infected, hospitalised and die of COVID-19 than white individuals.[15] Similarly, high morbidity and mortality rates were seen in ethnic minority groups in the UK.[16] This raises an important ethical question: should individuals from disproportionately

affected racial minorities be prioritised when allocating scarce resources such as ventilators and vaccines? If so, how should this be weighed against other ethical values in resource allocation, including the desire to save the most lives and the need to treat patients equally?

Some legal scholars, policy advocates and ethicists have suggested that race should be factored into ventilator and vaccine allocation.[17–19] For example, some have proposed positive discrimination in the form of equity weights.[12] However, such an approach might raise political[20] or legal concerns.[21] Alternatively, it could lead to ethical questions. Treating members of racial groups differently would conflict with principles of equality. Furthermore, in some circumstances, preferential allocation to members of a disadvantaged group may possibly increase overall mortality from COVID-19.[22 23] On the other hand, not considering it could widen the race-based difference in COVID-19 deaths and conflict with the principle of equity.

Policy discussion during the pandemic has largely lacked information on public opinion: it is not known what people in countries like the USA and UK, with high racial inequality in COVID-19 deaths, think about these questions. There is some evidence that the public support including race in allocation of vaccines,[24] but none about the *relative* importance of medical versus racial factors.

We sought to examine the relative weight given by the public to race in allocation and to compare this with participants' views on obesity and male sex. Race, obesity and sex are independent risk factors for COVID-19 (in one UK study, HRs for COVID-19 death were 1.48, 1.40 and 1.59, respectively),[25] but they differ ethically (eg, in their association with historical injustice). Finally, we sought to explore whether prioritisation preferences differed between racial groups, socioeconomic status or political orientations or were related to stereotypes and covert racial attitudes. This would illuminate what drives prioritisation attitudes but also whether a broader consensus exists.

## METHODS

The study was approved by the University of Oxford Central University Research Ethics Committee (R73841/RE002). It follows the American Association for Public Opinion Research (AAPOR) reporting guideline. All data, code and materials used in the analysis are available in a public, open access repository.[26]

Participants were recruited from large market research panels and paid US$10.8/hour. Speed and attention checks were used to identify and exclude respondents not paying sufficient attention to question details.

Two surveys were conducted in December 2020/January 2021 (USA n=1296, UK n=1284) (table 1). Each subsample size gave the ability to estimate true preferences with a 4% margin of error (95% CI) for the US and the UK population. The survey achieved a completion rate (completed surveys divided by number of respondents who entered the survey) of 86.7%. As the number of people that had the chance to participate in the opt-in panel is not known, we cannot report the AAPOR Response Rate (RR1).[9]

The surveys were adapted from a previous study.[27] Participants were asked to imagine that they needed to make decisions in the setting of a severe shortage of ventilators or vaccines. To examine how much weight participants would give to predicted chance (based on the age and comorbidity) of survival for allocating ventilators or of developing severe COVID-19 for vaccines, participants read scenarios where potential recipients differed in those variables to different degrees. For example, in one ventilator scenario, one patient had a 50% chance of survival and the other a 40% chance. In a corresponding vaccine scenario, participants had a 20% versus 10% chance of developing severe COVID-19. Participants decided whom to give the ventilator or vaccine to or to toss a coin (figure 1). Thereafter, participants were randomly introduced to one of the three additional factors: race, sex and obesity.

For a racial minority in the USA, we chose 'black recipients' and in the UK, 'BAME recipients'. BAME (black, Asian and minority ethnic) is a common term in the UK, used to describe non-white ethnic groups.[28] However, while commonly used, the term is no longer in official use as it combines ethnic groups with distinct identities.[29] Before completing scenarios, we reminded participants of the racial inequality in COVID-19-related deaths in the UK and in the USA. Thereafter, participants worked on five scenarios involving one black (BAME) and one white patient. In one scenario, both patients had the same chance of surviving/severe COVID-19; in second scenario the black (BAME) patient had a higher chance of survival/severe COVID-19; in both scenarios the white patient had a higher chance. Participants indicated for each whether they wanted to give the ventilator/vaccine to the patient from a racial minority, the white patient or toss a coin.

Participants saw a similar set of scenarios for sex and obesity as additional risk factors (see supplemental materials(SM)). To test whether it would make a difference, we repeated the scenarios for groups of patients.

We asked participants questions to capture their perceptions of racial minorities, men and obese people, including their views about the degree to which worse outcomes from COVID-19 were a result of social injustice or under the control of the individual. We also administered the Modern Racism Scale, intended to capture covert discriminatory attitudes.[30] The Modern Racism Scale is one of the most commonly used and best validated instruments to examine prejudice against black people in the USA.[31]

### Statistical analysis

To examine the differences between risk factors and the variables that impacted participants' attitudes, we created a composite attitude score for each set of scenarios by counting the number of times participants decided to

**Table 1** Demographic details of participants

| UK/USA | US sample n (%) | UK sample n (%) |
|---|---|---|
| Gender | | |
| Male | 634 (48.9) | 630 (49.1) |
| Female | 656 (50.6) | 650 (50.6) |
| Other | 3 (0.2) | 3 (0.2) |
| Prefer not to say | 3 (0.2) | 1 (0.1) |
| Race | | |
| White | 1034 (79.8) | 1103 (85.9) |
| Hispanic or Latino | 70 (5.4) | – |
| Black or African American/Black or Black British | 114 (8.8) | 41 (3.2) |
| Native American or American Indian | 19 (1.5) | – |
| Mixed | – | 29 (2.3) |
| Asian or Pacific Islander/Asian or Asian British | 23 (1.8) | 98 (7.6) |
| Other | 21 (1.6) | 11 (0.9) |
| Prefer not to say | 11 (0.8) | 2 (0.2) |
| Age | | |
| 18–24 | 159 (15.5) | 145 (11.3) |
| 25–34 | 248 (18.9) | 303 (23.6) |
| 35–44 | 237 (18.9) | 291 (22.7) |
| 45–54 | 252 (18.3) | 238 (18.5) |
| 55–64 | 224 (14.2) | 198 (15.4) |
| 65–74 | 129 (5.9) | 89 (6.9) |
| 75–84 | 38 (1.6) | 20 (1.6) |
| 85 or older | 4 (0.6) | – |
| Highest level of schooling | | |
| Less than high-school degree/primary school | 24 (1.9) | 3 (0.2) |
| High-school graduate or equivalent/secondary school up to 16 years | 237 (18.3) | 236 (18.4) |
| Some college but no degree/higher or secondary or further education | 250 (19.3) | 441 (34.3) |
| Associate degree in college (2 years) | 122 (9.4) | – |
| Bachelor's degree | 160 (12.3) | 381 29.7) |
| Master's degree | 290 (22.4) | 138 (10.7) |
| Doctoral degree | 29 (2.2) | 29 (2.3) |
| Professional degree (JD, MD) | 38 (2.9) | 47 (3.7) |
| Prefer not to say | 6 (0.5) | 9 (0.7) |
| Other | 140 (10.8) | – |
| Employment status | | |
| Employed part time | 136 (10.5) | 735 (57.2) |
| Employed full time | 617 (47.6) | 232 (18.1) |
| Unemployed looking for work | 96 (7.4) | 70 (5.5) |
| Unemployed not looking for work | 113 (8.7) | 96 (7.5) |
| Retired | 274 (21.1) | 114 (8.9) |
| Student | 48 (3.7) | 37 (2.9) |
| Disabled | 12 (0.9) | – |
| Estimated household income | | |
| $0–$25 k/ £0–£20 k | 230 (17) | 263 (20.5) |
| $25k–$50k/£20k–£30k | 274 (21.1) | 276 (21.5) |

**Table 1** Continued

| UK/USA | US sample n (%) | UK sample n (%) |
|---|---|---|
| £30k–£40k | – | 198 (15.4) |
| £40k–£50k | – | 145 (11.3) |
| $50k–$75k/£50k–£60k | 224 (17.3) | 106 (8.3) |
| $75k–$100k/£60k–£100k | 168 (13) | 191 (14.9) |
| $100k–$150k/£100k+ | 199 (15.4) | 105 (8.2) |
| $150k+ | 103 (7.9) | – |
| Prefer not to say | 11 (0.8) | – |

Participants were sampled to be representatives of the US and UK general population for sex, age, race, household income and employment. (Different income categories were used to reflect known information on population characteristics.) Fourteen per cent of the UK sample and 19% of the US sample described their racial background as non-white. Note that, compared with the last census,[46] the US sample had a lower proportion of Hispanic and black participants than the general US population.

give the ventilator or vaccine to the person or group with an additional risk factor in successive scenarios. To test whether differences existed between risk factors, we used repeated measure analysis of variance with Bonferroni corrected post hoc comparisons. To test which variables impacted participants' attitudes toward risk factors, we ran a series of four linear models to explore the role of various factors in predicting racial prioritisation attitudes. For each model, we defined attitudes as dependent variable and subjects as the random factor. In the first two models, we entered age, gender, race, relative income, educational level, political ideology (general, social and

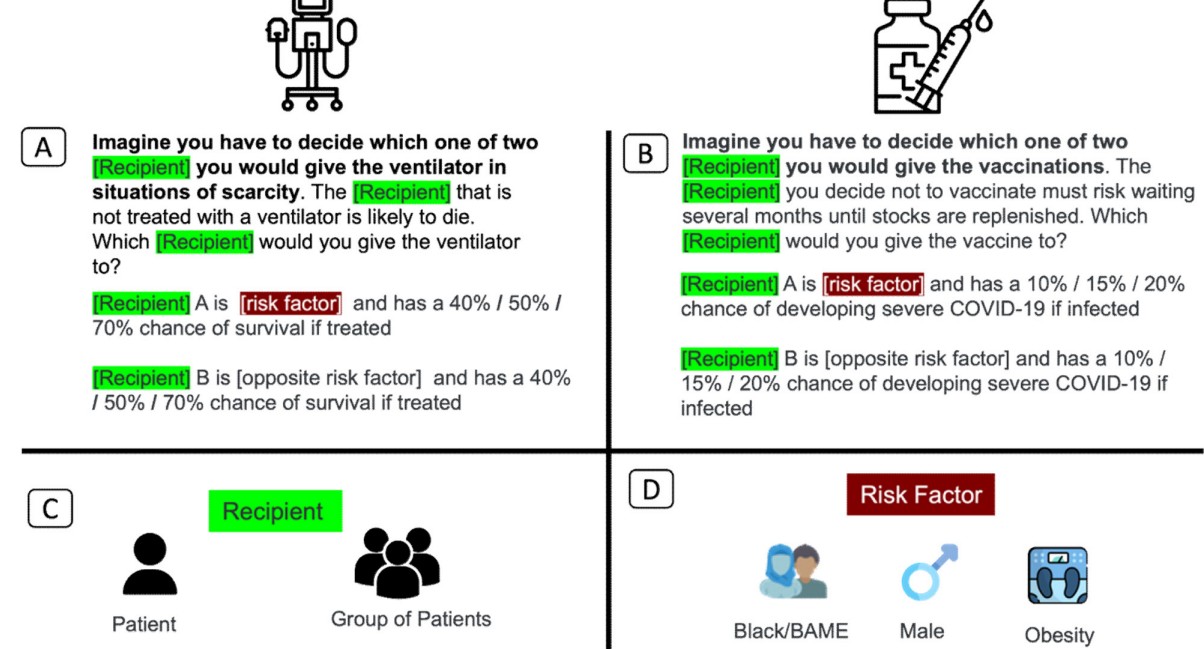

**Figure 1** Survey design. Participants from the USA (n=1296) or the UK (n=1284) answered questions about their preferences for allocating scarce medical resources (online sample, aiming to be nationally representative for sex, age, race, household income and employment—see online supplemental methods). Fourteen per cent of the UK sample and 19% of the US sample described their racial background as non-white. (A) Half the participants (n=1262) saw scenarios relating to ventilators. They had to decide which of two potential recipients should get the ventilator. Potential recipients had different chances of surviving COVID-19 with ventilation (40%, 50% or 70%) and one of them had an additional risk factor. Participants could give it to either one of the potential recipients or choose to toss a coin to decide. (B) The other half of participants (n=1318) saw scenarios relating to vaccines. Here, they had to decide which of two potential recipients should get the vaccine. Potential recipients had different chances of developing severe COVID-19 (10%, 15% or 20%), and one of them had an additional risk factor. Again, participants could give it to either one of the two potential recipients or toss a coin. (C) Potential recipients were either individual recipients or groups of recipients. (D) Additional risk factors used in our study were racial minority status (black in the USA, BAME in the UK), being male or being obese. The opposite additional risk factors were white, female and healthy weight, respectively. BAME, black, Asian and minority ethnic.

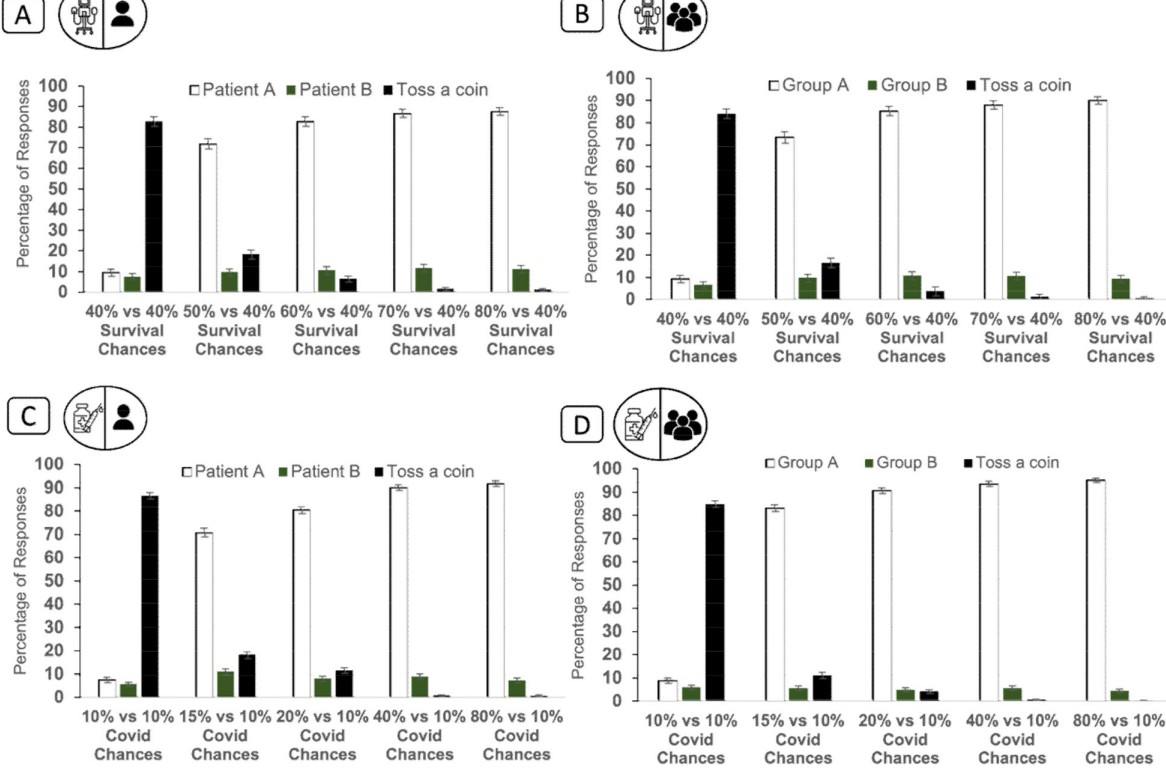

**Figure 2** The role of chances of developing severe COVID-19 for vaccine allocation and chances of survival for ventilator allocations in the absence of additional risk factors. Participants saw either scenarios that involved individual patients (A,C) or groups of patients (B,D). For both individual patients and groups of patients, the greater the difference in chances of survival, the more participants were willing to allocate the ventilator to the potential recipient more likely to survive (A,B). Similarly, the larger the difference in chances of developing severe COVID-19 if infected, the more participants were willing to allocate the vaccine to the potential recipient at higher risk (C,D) (error bars=95% CI). Note that, in these scenarios (involving only medical risk factors), we gave participants a wider range of outcome variation between recipients than in later scenarios involving additional risk factors. (For symbols, see figure 1).

economical) and type of scenario (patient vs group based) as fixed factors. In the second two models, we additionally added perceptions of injustice, responsibility, warmth, competence and modern racism scores. The full results of each model can be seen in the online supplemental results (online supplemental table 1). We used SPSS 28 for all analyses.

## Patient and public involvement
No patient involvement.

## RESULTS
### Prioritisation of medical risk
We first assessed the impact of medical risk factors on allocation. Asked to choose between patients 50% versus 40% chance of surviving if treated, 71.9% (95% CI: 69.3% to 74.3%) of participants gave the ventilator to the patient with higher chances. For vaccines (15% vs 10% chance of severe COVID-19), 70.8% (95% CI: 68.3% to 73.2%) gave the vaccine to the higher risk recipient. This increased when the difference between patients increased (figure 2). When there was no difference, 82.7 (86%) (95% CI: 80.5 to 84.8, 84.6 to 88.4) tossed a

coin. Patterns of response were very similar for scenarios involving groups as for individual patients (figure 2).

### Risk factors and resource allocation
In scenarios involving patients with identical chances of survival but different races, 14.8 of participants allocated the ventilator to black or BAME patients, 68.9% chose to toss a coin and 16.2% chose the white patient. For vaccine allocation to patients with identical chances of severe illness, 43.6% of participants allocated the vaccine to black or BAME patients, 49.2% chose to toss a coin and 7.2% chose the white patient. When the racial minority patient had a lower chance of survival or severe illness, <15% of respondents allocated the ventilator or vaccine to them (figure 3).

When chances of survival were equal, but one patient was male and the other female, 11.9% (95% CI: 10.2% to 13.8%) of participants gave the ventilator to the male, (22.7% female patient, 65.4% tossed a coin). When obesity was an additional risk factor, 11.9% (95% CI: 11.6% to 15.5%) chose the obese patient (41.1% healthy weight, 45.4% tossed a coin).

For vaccines, in scenarios of equal medical risk, where one patient was male and the other female, 44.2% of

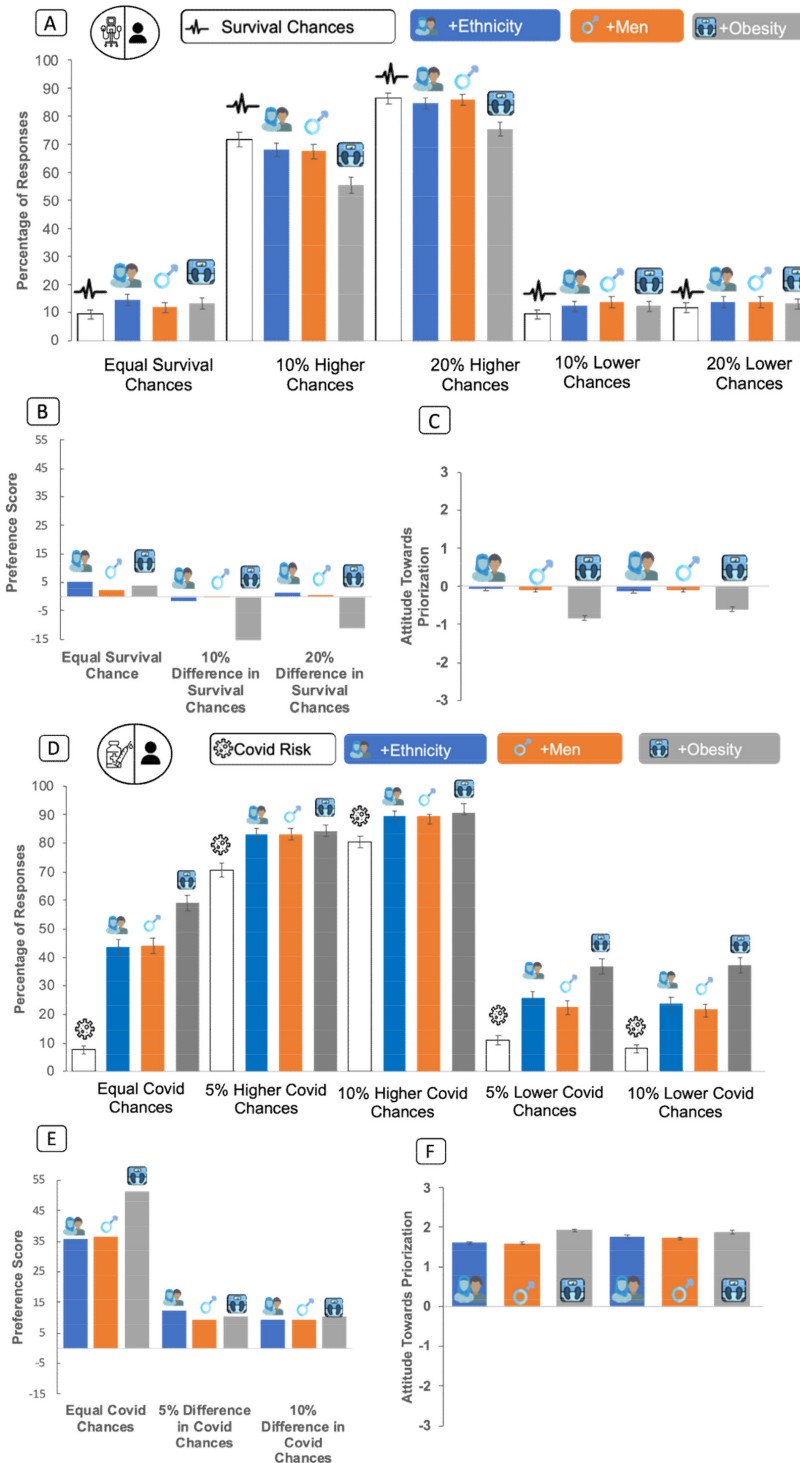

**Figure 3** (A–C) Preferences for and attitudes toward prioritisation of ventilators. (A) Percentages of respondents willing to give the ventilator to the racial minority/male/obese patient when both patients had either a similar or a 10% or 20% higher or lower survival chance. For comparison, we added the responses for scenarios without additional risk factors (white bars, labelled 'survival chances'). (B) For illustrative purposes, we subtracted the percentage of answers on the scenarios without additional risk factors from those with them. Race and sex did not elicit robust preferences, while obesity elicited negative prioritisation preferences. (C) Our composite attitude score mirrored the outlined findings. (D–F) Preferences for and attitudes toward prioritisation of vaccines. (D) Percentages of respondents willing to give the vaccine to the recipient with an additional risk factor when their chances of developing severe COVID-19 were either the same or were 5% or 10% higher or lower than the recipient without an additional risk factor. For comparison, we added the answers for scenarios without additional risk factors (white bars: labelled 'COVID-19 risk'). (E) For illustrative purposes, we subtracted the percentage of answers on the scenarios without additional risk factors from those that contain them. A clear positive preference for each factor emerged, with obesity eliciting the strongest positive responses. (F) Our composite attitude score mirrored the outlined finding. For vaccines, each factor elicited a positive preference, obesity being strongest (error bars=95% CI).

participants gave the vaccine to the male patient. When obesity was an additional risk factor, 59.2% allocated to an obese patient (figure 3).

Using the composite attitude score, race and sex did not affect allocation decisions for ventilators, but obesity did so negatively (p<0.001, 95% CI −0.9485 to −0.7124). Participants had lower prioritisation preferences for obese patients (compared with racial minority, p<0.001, and male patients, p=0.030). For vaccines, participants had significant prioritisation preference for race, sex and obesity (attitude greater than zero: p<0.00001; post hoc analyses: obese patients>racial minorities, p<0.001, and male patients, p<0.001.) There was a very small main effect for scenario type (group preference—online supplemental figure 1). Country of participants (US vs UK) did not alter allocation attitudes ($F(1,1260)=0.619$, p=0.431).

### Determinants of prioritisation

Using mixed-linear models (online supplemental table 1, full results), we found that race of participant impacted ventilator allocation attitudes in the UK ($F(4,602)=6.95$, p<0.001) but not in the USA ($F(6, 635)=1.38$, p=0.21). A sensitivity analysis showed that the US sample afforded us a power of 0.99 to find a small difference between white and black participants. In the UK, Asian and black participants had more positive attitudes toward race in allocation of ventilators than white participants (SM). Among BAME participants, 22.1% of participants gave the ventilator to a BAME patient with a 10% lower chance of survival, while only 6.9% of the white participants did (see table 2).

Age, gender, socioeconomic status, education and political ideology did not impact ventilator allocation attitudes. Those participants who strongly endorsed ventilator allocations based on chances of survival in the scenarios without any additional risk factors were not more or less likely to reject ventilator allocation based on race (r=−0.047).

Race of participants did not affect vaccine allocation. In the USA (but not the UK), gender ($F3,661)=7.238$, p<0.0001) and social political orientation $F(1,661)=11.388$, p=0.001) impacted vaccine allocation based on race. Male participants had a less positive attitude (p=0.003), and more socially conservative participants

gave less weight to race in vaccine allocation (parameter estimate=−0.305, 95% CI=−0.480 to −0.130).

### Influence of attitudes toward race on prioritisation

In both countries, participants were more likely to believe that adverse COVID-19-related health outcomes were the result of injustice and outside the individual's control for race than for obesity/male sex (online supplemental figure 2). Obesity was associated with the strongest sense of individuals' health outcomes being within their control. US and UK respondents had similar attitudes toward the respective racial minorities (SM).

In the USA and the UK, responses on the Modern Racism Scale were associated with attitudes toward ventilator and vaccine allocation based on race (ventilator: USA: $F(1,634)=10.73$, p=0.001, UK: $F(1,624)=13.88$, p<0.0001; vaccine: USA: $F(1,621=23.00$, p<0.0001, UK: $F(1,607)=16.98$, p<0.0001). The more participants endorsed modern racism statements, the less positive their composite attitudes toward ventilator or vaccine allocation to racial minority patients. A minority (27.3%) of participants had a mean score of higher than 3 on the Modern Racism Scale, indicating an endorsement of modern racism. These participants did not give weight to race in ventilator or vaccine allocations (online supplemental figure 3). Participants with low modern racism scores also did not give weight to race in ventilator allocation but did for vaccines. Modern racism did not affect prioritisation based on sex/obesity (online supplemental figure 3). Perceptions of injustice did not predict ventilator or vaccine allocation attitudes in the USA ($F(1,634)=0.778$, p=0.378) or the UK ($F(1,624)=2.732$, p=0.099).

### DISCUSSION

Previous research has identified some of the public's prioritisation preferences in relation to race and vaccines.[24] To our knowledge, this is the first study to examine the views of the public on race in ventilator allocation and in comparison with other factors. This international study was conducted during the second wave of the COVID-19 pandemic in two countries with documented and publicised adverse outcomes for racial minorities. Survey participants largely allocated ventilators and vaccines on

**Table 2** Differences in responses between UK white and BAME participants for all scenarios involving ventilators and race

| | Ethnic minority participants | | | White participants | | |
|---|---|---|---|---|---|---|
| | BAME patient | Coin toss | White patient | BAME patient | Coin toss | White patient |
| Equal survival chances | 24.4 | 72.1 | 3.5 | 13 | 73.2 | 13.8 |
| 10% higher chance | 77.9 | 11.6 | 10.5 | 73.2 | 15.3 | 11.5 |
| 30% higher chance | 87.2 | 10.5 | 2.3 | 90.5 | 2 | 7.4 |
| 10% lower chance | 22.1 | 15.1 | 62.8 | 6.9 | 15.5 | 77.7 |
| 30% lower chance | 4.7 | 22.1 | 73.3 | 1.5 | 8 | 90.5 |

BAME, black, Asian and minority ethnic.

the basis of medical factors and independent of the race of recipients. They were highly sensitive to small differences in predicted outcome. For example, for a 10% difference in predicted survival, 71.9% of participants gave the ventilator to the patient with higher chances while 82.7% chose a coin toss where there was no difference in predicted chance. For vaccine allocation, we found similar results for even smaller differences in risk of severe COVID-19 (5% difference). (It is important to note that such small differences in prognosis are clinically impossible to achieve but demonstrate participants' sensitivity to medical factors.)

Relatively few respondents in our survey, even those from racial minorities themselves, endorsed priority for ventilators for those from racial minorities. However, BAME respondents in the UK (but not black respondents in the USA) were more inclined than white respondents to preferentially allocate a ventilator to a patient from a racial minority. For vaccines, where other factors were equal, 40%–60% of respondents allocated preferentially to patients/groups with additional risk based on race, sex or obesity. In the survey, willingness to prioritise based on additional risk did not correlate with participants' perceptions of whether risk factors were caused by social injustice. However, covert racial attitudes (endorsed by approximately a quarter of respondents) negatively impacted attitudes to allocation in both countries.

The different weights given to race in the allocation of vaccines versus ventilators in our study may reflect people's desire to prevent the most deaths.[7 32] Previous research has shown strong public endorsement for prioritising chances of survival in ventilator allocation and probability of severe illness for vaccine allocation.[24 27 33–35] However, whereas vaccines are provided to prevent illness, ventilators are provided to those already critically ill. Participants might have felt that allocating ventilators preferentially to those with additional risk factors (race, sex, obesity) could lead to more deaths from COVID-19 if such patients (even with otherwise equal medical risk) have higher mortality rates. Another possible explanation, at least for US participants, is that some were aware of more recent evidence of similar hospitalised critical care outcomes for black patients.[36] This may have led some to conclude that since hospital admissions were in proportion to the occurrence of severe disease in the community, allocation of critical care resources preferentially would not be justified.[37] (Since the survey was conducted prior to much of this evidence emerging, this was potentially less likely to influence responses). Respondents might have regarded preferential allocation of a life-saving intervention like ventilators as more legally or ethically questionable than prioritisation of vaccines. Finally, participants might have reasoned that ventilators do not have wider community benefits, but vaccines do. Hence, preferential allocation of vaccines represents a more substantial (and possibly less controversial) means to reduce inequality.

However, in our study participants who expressed a strong attitude for ventilator allocation based on chances of survival were not more or less likely to reject preference in ventilator allocation based on race. Other ethical principles or considerations may be guiding allocation decisions, for example, concern for equality and avoiding discrimination.[38 39] Respondents did not believe that racial minorities should be at a disadvantage for access to ventilators—reinforcing the importance of ensuring that medical criteria used for allocation are not racially biased.[40 41]

One reason to potentially give priority in allocation of scarce treatment to racial minorities is because of concerns over past or present injustice.[12] In our survey, UK and US respondents perceived higher death rates among minorities to be more unjust and less under individual control than the mortality rates of male and obese recipients. However, these perceptions did not affect allocation decisions. Similarly, political orientation had little impact on allocation decisions in our survey. More conservative respondents were less inclined to indicate that injustice is responsible for higher death rates. However, this perception did not affect allocation decisions.

Allocation decisions in our survey were associated with responses on the Modern Racism Scale. This instrument is intended to capture covert racial attitudes.[30] While people who score high on the Modern Racism Scale potentially do not perceive themselves as racist, discriminatory attitudes can nevertheless guide their decisions. Modern racism might measure negative attitude toward affirmative action policies in general. However, in our survey, modern racism did not predict attitudes to sex or obesity in allocation. Furthermore, in our models, political ideology was not associated with allocation based on race. Taken together, these findings suggest that some opposition to giving weight to race in prioritisation arises from covert racial attitudes. However, even those respondents with low scores on the Modern Racism Scale (three-quarters of those surveyed) did not on average give weight to race in allocation of ventilators. Covert racial attitudes do not appear to explain our overall finding of the lack of weight given to race in public attitudes to ventilator allocation.

Importantly, our results help contextualise previous findings that looked at race in allocation in isolation. For example, a US Gallup/COVID-19 Collaborative survey found that 74%–85% respondents supported giving priority for vaccine access to black, Hispanic and native Americans.[24] In our study, slightly less than half of respondents endorsed this, but only in cases where other risk factors were equal. This highlights the importance of clarifying the weight given to different ethical values when surveying the public's views on resource allocation but also points to the importance of alternative approaches to lessening the impact of COVID-19 on racial minorities. One promising future direction would be to use computational models of decision-making to quantify the weight people give in their allocation decision to different medical and social factors when presented simultaneously, as is the case in clinical practice.

## Strengths and limitations

The hypothetical scenarios used in our study enabled us to assess and compare the attitude of the public toward allocation of two key resources in a pandemic. The use of parallel examples enabled appraisal of the relative ethical weight given to some important risk factors compared with the risk of serious illness or death. Scenarios were highly specific, somewhat artificial and clinically unrealistic (particularly in relation to small predicted differences between patients or groups). This could have differed from participants' lived experience and may have contributed to unease in prioritisation decisions. The survey design did not follow an explicit social justice framework.[42 43]

Survey responses allowed us to quantify some elements of the public's values, but we were not able to assess the reasons why particular answers were given. Future research should explore this.

The survey was conducted during the COVID-19 pandemic at a fraught and emotional time. This would have increased the salience of the survey for participants and could provide a relevant snapshot of community values at that point in time. However, responses might have been different if the survey had been conducted at a different time. For example, respondents might have been sensitive to the controversial nature of the topic. They may have been more likely to advocate for correcting injustices if the survey were conducted outside a pandemic.

One limitation is that despite our attempt to obtain a nationally representative sample, in our US sample, Hispanic and black participants were under-represented. Thus, our study might underestimate how different racial groups feel about allocation of scarce resources. Furthermore, although we aimed to create and word questions in a neutral and inclusive way, the involvement of black and BAME community members would have been beneficial in the survey design.

Another limitation is the lack of involvement of non-academic members of the black and BAME community. Recent frameworks of survey development encourage active involvement of racial minorities in the development and analysis of research as best practice to ensure, for instance, that the lived experiences of community members are represented when designing the survey and that different perspectives of how the same data could be interpreted are considered.[44] Hence, it is important to acknowledge that our questions, data collection and interpretation only represent one perspective and that involvement of non-academic black and BAME community members might have produced different results.

## CONCLUSION

Obviously, the results of this survey do not settle the ethical question of whether governments should prioritise based on race or other forms of social disadvantage. Ethical argument and analysis may lead to prioritisation policies that differ from those endorsed by the majority. However, understanding the general approach of the public toward prioritisation in settings of severe scarcity may be important in debate and deliberation: it may challenge or confirm assumptions about what the public would or would not support, as well as highlight areas of ethical consensus or point to important areas of disagreement.[45] Our survey identified that in specific allocation scenarios involving high demand and severe resource shortage, a majority of US and UK respondents chose to allocate ventilators to patients with predicted higher chance of survival and vaccines to those at higher risk of severe illness, regardless of patient race. A very small proportion of respondents (<15%) prioritised ventilator allocation to patients from a racial minority, though this was higher among UK respondents from ethnic minority backgrounds. We also found support from a large minority for inclusion of race as a tie-break consideration in vaccine allocation. Although we may hope that it is never needed, these findings may be relevant to planning for the next pandemic.

**Contributors** AK, HZ and DW designed the survey. JS, IS and WS-A provided crucial feedback on methodology and design. HZ organised the collection of the data. AK analysed the data. AK, HZ and DW wrote the manuscript. JS, IS and WS-A provided crucial feedback. DW is the guarantor.

**Funding** This study (survey costs) was commissioned and paid for by the WHO (2020/1077166). Copyright in the original work on which this article is based belongs to WHO. The authors have been given permission to publish this article. The authors alone are responsible for the views expressed in this publication and they do not necessarily represent the views, decisions or policies of the WHO. Researchers were funded in part, by the Wellcome Trust (203132/Z/16/Z) (DW and HZ) and by the Arts and Humanities Research Council (AHRC) (DW) as part of the UK Research and Innovation rapid response to COVID-19 (AH/V013947/1), as well as the NIHR Oxford Health Biomedical Research Centre (IS-BRC-1215-20005) (IS). The funders had no role in the preparation of this manuscript or the decision to submit for publication. For the purpose of open access, the authors have applied a CC BY public copyright license to any Author Accepted Manuscript version arising from this submission.

**Competing interests** None declared.

**Patient and public involvement** Patients and/or the public were not involved in the design, conduct, reporting or dissemination plans of this research.

**Ethics approval** This study involves human participants. The study was approved by the University of Oxford Central University Research Ethics Committee (R73841/RE002). Participants gave informed consent to participate in the study before taking part.

**Provenance and peer review** Not commissioned; externally peer reviewed.

**Data availability statement** Data are available in a public, open access repository. All data, code and materials used in the analysis can be found at: osf.io/mhyqz.

**ORCID iDs**
Hazem Zohny http://orcid.org/0000-0002-7734-2186
Dominic Wilkinson http://orcid.org/0000-0003-3958-8633

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
