## [Reviewer comments · BMJ Open]

ARTICLE DETAILS

TITLE (PROVISIONAL)	Race and resource allocation: An online survey of US and UK adults' attitudes towards COVID-19 ventilator and vaccine distribution
AUTHORS	Kappes, Andreas; Zohny, Hazem; Savulescu, Julian; Singh, Ilna; Sinnott-Armstrong, Walter; Wilkinson, Dominic

VERSION 1 – REVIEW

REVIEWER	Hick , JL University of Minnesota Hennepin-University Partnership, Emergency Medicine
REVIEW RETURNED	22-Apr-2022

GENERAL COMMENTS	Thank you for the opportunity to review this interesting, timely, and relevant submission. The question of how race should be incorporated into scarce resource allocation is pressing and this article contributes to our understanding of some the influences and situations that are relevant to the discussion. Please find below my comments and thoughts on improving your manuscript. 1) I found the use in the abstract of the parentheses to represent the vaccine data to be a bit confusing. Consider separating the data or perhaps use italics (for the word vaccine and subsequent data points) to further clarify the distinction in results? 2) It is unlikely, but possible that your survey population was aware that though there is substantial difference in incidence and deaths of non-majority population members from COVID-19 that there is no difference in hospitalized critical care outcomes that has been demonstrated for blacks (consider referencing https://jamanetwork.com/journals/jama-health-forum/fullarticle/2787469?utm_source=silverchair&utm_campaign=jama_network&utm_content=covid_weekly_highlights&utm_medium=email table 2 or https://jamanetwork.com/journals/jamanetworkopen/fullarticle/2785980) - thus I and some others have argued that since admissions to the hospital are in proportion to the occurrence of severe disease in the community that allocation of critical care resources preferentially cannot be justified on the basis of COVID outcomes (https://www.thelancet.com/journals/eclinm/article/PIIS2589-5370(21)00118-8/fulltext) in contrast to vaccine which clearly has community benefit when given to those most likely to contract / spread / and have complications from the disease. This type of argument could have influenced the respondent's decisions and may be worth discussion 3) It may have been more helpful to give clinically similar prognosis and ask respondents to determine if they would use race as a 'tie-breaker' as this is much more common than weighing specific, small differences in prognosis. From a clinical standpoint, a 10-20% difference (in one point in the text it mentions a 60/40 outcome difference, in the methods it mentions a 10% difference?) is meaningless - it's doubtful that the respondents would understand this but the narrow margins may have contributed to unease about allocating ventilators given the marginal difference between the patients. This may be worth comment as well. 4) Can you comment further in the manuscript about the validation of the racism assessment instrument that was used and any alternatives that were considered? How are the responses benchmarked to yield an association with the individual's views?
---

	5) Gostin's article is cited, but it may be worth additional comment on the legal difference between vaccine allocation and ventilator allocation that could have driven additional discomfort with ventilator allocation. Vaccine allocation to at-risk populations is fully defensible from a public health standpoint as control of the disease in these populations has overall benefit for society as well as the individuals. Conversely, allocation of a ventilator for reasons other than prognosis directly harms individuals that did not receive it - in the United States at least this has significant legal implications and indeed, multiple lawsuits have been filed challenging allocation of treatments designed to prevent hospitalization to higher-risk populations which involve far less harm than ventilator deprivation (https://www.washingtonpost.com/health/2022/02/10/conservatives-covid-treatments-race/). The legality of including factors aside from clear prognostic evidence for better short-term survival has not been directly addressed by the courts, but to include racial, economic, or other variables would likely be problematic, no matter how well intended. 6) Finally, this research was conducted during the COVID-19 pandemic at a highly fraught and emotional time - it is possible that the results of the survey would be different if they occurred in a non-pandemic timeframe (e.g. that respondents may have been more likely to advocate correction of historical inequity) - this also may be worth comment In the end, translating ethical ideals into just operational allocation schemes can be elusive. This paper does an excellent job illustrating that there is likely significant room for community consensus on allocation of vaccine but that allocation of critical care resources on the basis of social factors is not agreeable to most citizens, even those belonging to impacted groups (albeit with small numbers represented). Structural inequity is not easily corrected at the bedside - we owe our non-majority communities better and targeted communications, access to care, access to prevention, diagnostics, and equal access to treatment modalities. Thank you for contributing to our understanding of the community's attitudes to racial prioritization of resource allocation.
--	---

REVIEWER	Bruce , Lori Interdisciplinary Center for Bioethics, Yale University, New Haven
REVIEW RETURNED	04-May-2022

GENERAL COMMENTS	Manuscript Review This non-blinded review considers Kappes et al's "Public Attitudes Towards Race and Resource Allocation in a Pandemic," submitted to BMJ Open. The author's survey asked members of the public whether vulnerable racial minorities ought to receive preferential weighting for vaccines and ventilators. Given the troubling and extensive racial disparities exhibited during the COVID-19 pandemic, this topic is timely and relevant. This review provides a short list of considerations related to the survey's design and analysis. Survey Design and Analysis Disability advocates often challenge physicians and policymakers with a mantra they adopted from the central European political tradition, "Nothing about us without us." This plea for inclusion speaks to the uninformed and sometimes harmful practices, policies, and laws enacted by the able-bodied to address healthcare needs of people with disabilities. When inclusion is present in disability policy formulation, care becomes more empathic and congruent with patient values. Good policy requires deep understanding of the issue at hand, and to acquire that understanding we need open, inclusive dialogue with the affected communities - especially vulnerable populations - from formulation through evaluation. Similarly, surveys on policies impacting vulnerable populations require inclusion. To uncover and report the views and beliefs of members of vulnerable racial groups on race and allocation in a
--

pandemic, surveys ought to be designed and interpreted with inclusion and representation from Black and BAME communities. The guidance outlined in this review is perhaps best articulated in the form of (1) A Toolkit for Centering Racial Equity Throughout Data Integration (developed at the Actionable Intelligence for Social Policy (AISP) at the University of Pennsylvania) and (2) Cornelius and Harrington's text, A Social Justice Approach to Survey Design and Analysis. These frameworks outline a series of recommendations to avoid reinforcing "legacies of racist policies and produc[ing] inequitable resource allocation."

Similarly, a trauma-informed lens seeks to explore what's "going on with" people instead of what is wrong with them. This view encourages open-ended inquiry and framing of issues through inclusion to more fully understand the nature of a problem.

Surveys on hospital policy can certainly integrate trauma-informed principles to more fully tap into patient experience.

The following are a few examples of how the AISP framework - and Cornelius and Harrington's guidance - could have produced a more informed and generalizable survey.

1. The AISP framework encourages "Clearly discerning who decides how to frame the problem."

Recent descriptions of community preferences for resource allocation in a pandemic by non-Black scholars have failed to accurately "frame the problem" by glossing over the preferences of Black communities. For instance, White and Lo, when citing Biddison's 2018 study, recently claimed:

Most people would agree that patients with an excellent prognosis for survival with treatment should be given priority over those with a poor prognosis on the grounds that, all other things being equal, it is desirable to save more lives rather than fewer.

This is a selective statement. While it is true that most in Biddison's study did prefer to maximize lives saved, Biddison broke down preferences by race and clearly noted that the majority of her Black participants preferred first-come-first-served (FCFS):

African American participants had significantly lower odds of wanting to always or often use "saving the most life-years" as a criterion for allocation decisions than their white colleagues (OR, 0.34; 95% CI, 0.21-0.58). Conversely, African American participants were significantly more likely to favor often or always using "first come, first served" to drive these key decisions (OR, 2.36; 95% CI, 1.29-4.29).

Given White and Lo's selective reporting, whenever ethicists frame the views of the public, we should use caution with any extrapolation or summarization especially if the views or experiences of white respondents are dissimilar to – and may drown out - the views of Black or BAME respondents.

In this light, I consider how Kappes et al framed their survey questions and the generalizability of their results. Kappes et al report that "participant race did not impact vaccine or ventilator allocation decisions in the US but did impact ventilator allocation attitudes in the UK ($F(4,602) = 6.95, P < 0.001$)."

In their conclusion, the authors state, "Our survey identified a broad consensus, in both the US and UK, for how scarce resources should be allocated during a pandemic." They also state, "...the

vast majority of participants (including those from minority groups) think that in situations of extreme scarcity, life-saving treatment like ventilators should be allocated in a race-neutral way... [a] finding [which] may be relevant to planning for the next pandemic." These conclusions seem to extrapolate too far.

The survey questions asked whether racial preference ought to be considered under quite specific scenarios. Their questions on ventilator allocation describe scenarios in which two patients need a ventilator and, in that moment, the survey participant needs to choose one patient. These are certainly important questions, but the findings may not more broadly infer that those respondents believe in race neutrality or condone the described allocation system.

Any number of reasons may account for participants who may decline to give preferential treatment within Kappes et al's survey but still disagree with the authors' stance of any kind of broad consensus for scarce resource allocation during a pandemic. For instance:

- Members of vulnerable racial groups may decline to give preferential treatment because racial weighting is controversial and may cause public uproar, while only having limited impact on correcting bias and racism within medicine.
- The survey spoke in conclusive terms that may not reflect lived experience – and as such would not be as generalizable. Perhaps, for example, there may be a distinction between responses to the survey's question (patients who have, for example, a 40% chance of survival) and a question which could have situated the scenario in a more lifelike and contextual frame (e.g., "a white physician declares a Black or BAME patient has a 40% chance of survival"). Patients (and perhaps especially those from vulnerable populations) often find physicians' perception of outcomes to be pessimistic, and physician prognosis is imprecise and may be tainted by bias and racism. The survey question, as written, does not reflect the nuance of how these scenarios present themselves, and as such, has limited relevance to future pandemic planning.

2. The AISP framework encourages "Clearly discerning who decides how to... determine what questions to ask." The survey's relevance would have been more pronounced if Black and BAME scholars – and community members – participated in crafting the survey questions (and interpreting its results). Perhaps, for instance, Black and BAME community members may have encouraged support for FCFS protocols, as was found in Biddison's study.

Similarly, Cornelius and Harrington recommend a community-inclusive survey development process which may include advance reviews by "Community-based ethics review boards." Perhaps such a review may have mitigated the US sample's underrepresentation from Black and Hispanic participants which (in Kappes' et al's words) may "underestimate how different racial groups feel about allocation of scarce resources."

3. The AISP framework encourages "Including qualitative stories to contextualize quantitative data." The authors' survey was limited in terms of the feedback it collected from participants. It did

	not ask participants why they made their choices. Free-form text is more cumbersome to code and analyze, and perhaps more costly, but it would have provided context and yielded insights on participants' preferences. In closing, Kappes et al's paper provides limited contributions to our understanding of public views on race and allocation in a pandemic. The survey design and analysis were not informed by a social justice paradigm, a racial equity lens, or a trauma-informed approach and as such their survey findings have limitations in terms of generalizability and insight. As Cornelius and Harrington state, when we tell "the story of barriers encountered by (vulnerable) populations," we are behooved to navigate with care. Actionable Intelligence for Social Policy. "A Toolkit for Centering Racial Equity Throughout Data Integration." University of Pennsylvania. https://aisp.upenn.edu/centering-equity/. Last accessed May 3, 2022. Lanphier, Elizabeth, and Uchenna E. Anani. "Trauma informed ethics consultation." The American Journal of Bioethics (2021): 1-13. Biddison ELD, Gwon HS, Schoch-Spana M, et al. Scarce resource allocation during disasters: a mixed-method community engagement study. Chest. 2018;153(1):187-195. doi:10.1016/j.chest.2017.08.001 White, Douglas B., and Bernard Lo. "Mitigating inequities and saving lives with ICU triage during the COVID-19 pandemic." American Journal of Respiratory and Critical Care Medicine 203, no. 3 (2021): 287-295. Biddison et al, 192. Cornelius, Llewellyn Joseph, and Donna Harrington. A social justice approach to survey design and analysis. Pocket Guide to Social Work Re, 2014. See p. 118.
--	--

VERSION 1 – AUTHOR RESPONSE

Reviewer: 1

Dr. JL Hick , University of Minnesota Hennepin-University Partnership

Comments to the Author:

Thank you for the opportunity to review this interesting, timely, and relevant submission. The question of how race should be incorporated into scarce resource allocation is pressing and this article contributes to our understanding of some the influences and situations that are relevant to the discussion.

Response:

Thank you for this very positive comment on our paper.

Please find below my comments and thoughts on improving your manuscript.

R1.1 I found the use in the abstract of the parentheses to represent the vaccine data to be a bit

confusing. Consider separating the data or perhaps use italics (for the word vaccine and subsequent data points) to further clarify the distinction in results?

Response:

Thank you. We have amended the abstract now, describing separately the vaccine and ventilator data:

Results: When asked to allocate ventilators to a racial minority or white patient with identical chances of survival, 14.8% allocated to the minority patient, 16.2% chose the white patient, and 68.9% chose to toss a coin. When the racial minority patient had a 10% lower chance of survival, 12.4% participants allocated the ventilator to the minority patient. When asked to allocate vaccines to a racial minority or white patient with identical risk of severe COVID-19, 43.6% allocated to the minority patient, 7.2% chose the white patient, and 49.2% chose a coin toss. When the racial minority patient had a 10% lower risk of severe COVID-19, 23.7% participants allocated the vaccine to the minority patient.

We have also changed the corresponding section in the results where we had used the same shorthand.

R1.2 It is unlikely, but possible that your survey population was aware that though there is substantial difference in incidence and deaths of non-majority population members from COVID-19 that there is no difference in hospitalized critical care outcomes that has been demonstrated for blacks (consider referencing https://jamanetwork.com/journals/jama-health-forum/fullarticle/2787469?utm_source=silverchair&utm_campaign=jama_network&utm_content=covid_weekly_highlights&utm_medium=email table 2 or <https://jamanetwork.com/journals/jamanetworkopen/fullarticle/2785980>) - thus I and some others have argued that since admissions to the hospital are in proportion to the occurrence of severe disease in the community that allocation of critical care resources preferentially cannot be justified on the basis of COVID outcomes ([https://www.thelancet.com/journals/eclinm/article/PIIS2589-5370\(21\)00118-8/fulltext](https://www.thelancet.com/journals/eclinm/article/PIIS2589-5370(21)00118-8/fulltext)) in contrast to vaccine which clearly has community benefit when given to those most likely to contract / spread / and have complications from the disease. This type of argument could have influenced the respondent's decisions and may be worth discussion

Response 1.2:

Thank you for pointing out this possible explanation. We have amended the text on page 9 to reflect it:

“The different weight given to race in the allocation of vaccines versus ventilators in our study may reflect people’s desire to prevent the most deaths.^{7,30} Previous research has shown strong public endorsement for prioritising chances of survival in ventilator allocation and probability of severe illness for vaccine allocation.^{24,27,31–33} However, whereas vaccines are provided to prevent illness, ventilators are provided to those already critically ill. Participants might have felt that allocating ventilators preferentially to those with additional risk factors (race, sex, obesity) could lead to more deaths from COVID-19 if such patients (even with otherwise equal medical risk) have higher mortality rates. Another possible explanation, at least for US participants, is that some were aware of more recent evidence of similar hospitalised critical care outcomes for Black patients.³⁵ This may have led some to conclude that since hospital admissions were in proportion to the occurrence of severe disease in the community, allocation of critical care resources preferentially would not be justified.³⁶ (Since the survey was conducted prior to much of this evidence emerging, this was potentially less likely to influence responses). Respondents might have regarded preferential allocation of a life-saving intervention like ventilators as more legally or ethically questionable than prioritisation of vaccines. Finally, participants might have reasoned that ventilators do not have wider community benefits, but vaccines do. Hence, preferential allocation of vaccines represents a more substantial (and possibly less controversial) means to reduce inequality.”

R1.3 It may have been more helpful to give clinically similar prognosis and ask respondents to determine if they would use race as a ‘tie-breaker’ as this is much more common than weighing specific, small differences in prognosis. From a clinical standpoint, a 10-20% difference (in one point in the text it mentions a 60/40 outcome difference, in the methods it mentions a 10% difference?) is meaningless – it’s doubtful that the respondents would understand this but the narrow margins may have contributed to unease about allocating ventilators given the marginal difference between the patients. This may be worth comment as well.

Response 1.3:

Thank you for this question. We appreciate that the specific numerical predictions of survival or severe COVID used in our survey are clinically unrealistic, since in practice fine-grained and accurate predictions are not available. However, to understand the views of the general public about the ethical weight given to prognosis (versus other factors), and to avoid the complication of uncertainty, we chose to provide specific numerical predictions.

In our survey we did seek to understand whether respondents would use race (or other factors) as a tie breaker. We stipulated in some scenarios that recipients/patients had identical predicted risk.

Page 6:

“In scenarios involving patients with identical chances of survival (severe illness) but different race, 14.8 (43.6%) of participants allocated the ventilator (vaccine) to Black or BAME patients, 68.9 (49.2%) chose to toss a coin, and 16.2 (7.2%) chose the White patient.

...

When chances of survival were equal, but one patient was male and the other female, 11.9% (95% CI: 10.2%,13.8%) of participants gave the ventilator to the male, (22.7% female patient; 65.4% tossed a coin). When obesity was an additional risk factor, 11.9% (95% CI: 11.6% - 15.5%) chose the obese patient, (41.1% healthy weight, 45.4% tossed a coin).”

We noted in the conclusion that a substantial proportion of respondents were willing to use race as a tie-break consideration for vaccines but not ventilators.

Page 11:

“However, we did find support from a large minority for inclusion of race as a tie-break consideration in vaccine allocation. We also found that the vast majority of participants (including those from minority groups) think that in situations of extreme scarcity, life-saving treatment like ventilators should be allocated in a race-neutral way, and preferentially to those with the highest chances of survival”

We agree with the reviewer that small differences in outcome might have led to unease about preferential allocation. In fact, we observed that respondents were highly sensitive to small differences in risk. We have added the following to the discussion. We have also added the reviewer’s concern to the limitations section (see E5 response).

Page 8:

“Survey participants largely allocated ventilators and vaccines on the basis of medical factors and independent of the race of recipients. They were highly sensitive to small differences in predicted outcome. For example, for a 10% difference in predicted survival, 71.9% of participants gave the ventilator to the patient with higher chances while 82.7% chose to toss a coin toss where there was no difference in predicted chance. For vaccine allocation we found similar results for even smaller differences in risk of severe Covid (5% difference). Few respondents in our survey, even those from racial minorities themselves, endorsed priority for ventilators for those from racial minorities.”

R1.4 Can you comment further in the manuscript about the validation of the racism assessment instrument that was used and any alternatives that were considered? How are the responses benchmarked to yield an association with the individual's views?

Response: Thank you. The Modern Racism Scale is one of the most widely used instruments to measure covert racial attitudes with more than 3600 citations for the scale publication. It measures covert racial attitudes in contrast to blatant racial attitudes and is the most commonly used and best validated instrument to examine prejudice against Black people in the US. We have added a sentence in the Methods section and an additional reference to highlight this fact. To form a score, the items of the scale are simply averaged, and a mean score higher than the midpoint of the scale indicates covert racial attitudes.

“We asked participants questions to capture their perceptions of racial minorities, men, and obese people, including their views about the degree to which worse outcomes from COVID-19 were a result of social injustice or under the control of the individual. We also administered the modern racism scale, intended to capture subtle or covert discriminatory attitudes.²⁹ The modern racism scale is one of the most commonly used and best validated instruments to examine prejudice against black people in the US.³⁰

R1.5 Gostin's article is cited, but it may be worth additional comment on the legal difference between vaccine allocation and ventilator allocation that could have driven additional discomfort with ventilator allocation. Vaccine allocation to at-risk populations is fully defensible from a public health standpoint as control of the disease in these populations has overall benefit for society as well as the individuals. Conversely, allocation of a ventilator for reasons other than prognosis directly harms individuals that did not receive it - in the United States at least this has significant legal implications and indeed, multiple lawsuits have been filed challenging allocation of treatments designed to prevent hospitalization to higher-risk populations which involve far less harm than ventilator deprivation (<https://www.washingtonpost.com/health/2022/02/10/conservatives-covid-treatments-race/>). The legality of including factors aside from clear prognostic evidence for better short-term survival has not been directly addressed by the courts, but to include racial, economic, or other variables would likely be problematic, no matter how well intended.

Response:

Thank you for noting another potential explanation for the differential response. Schmidt and Gostin's article points out the legal problem with the explicit use of race for policies relating to vaccine (or potentially equally to ventilator) allocation. We agree with the reviewer that particular sensitivity about the legality of ventilator prioritisation might have influenced responses and we have added this to the discussion.

Page 9:

“Respondents might have regarded preferential allocation of a life-saving intervention like ventilators as more legally or ethically questionable than prioritisation of vaccines.”

R1.6 Finally, this research was conducted during the COVID-19 pandemic at a highly fraught and emotional time - it is possible that the results of the survey would be different if they occurred in a non-pandemic timeframe (e.g. that respondents may have been more likely to advocate correction of historical inequity) - this also may be worth comment.

Response:

Thank you for pointing this out. We now raise this point as well in the new sub-section on the study's limitations. See response E5.

Reviewer: 2

Prof. Lori Bruce, Interdisciplinary Center for Bioethics, Yale University, New Haven

Comments to the Author:

Please kindly see the attachment

This non-blinded review considers Kappes et al's "Public Attitudes Towards Race and Resource Allocation in a Pandemic," submitted to BMJ Open. The author's survey asked members of the public whether vulnerable racial minorities ought to receive

preferential weighting for vaccines and ventilators. Given the troubling and extensive racial disparities exhibited during the COVID-19 pandemic, this topic is timely and relevant. This review provides a short list of considerations related to the survey's design and analysis.

Response

We are very grateful to Prof Bruce for these thoughtful and constructive comments on our paper.

Survey Design and Analysis

Disability advocates often challenge physicians and policymakers with a mantra they adopted from the central European political tradition, "Nothing about us without us." This plea for inclusion speaks to the uninformed and sometimes harmful practices, policies, and laws enacted by the able-bodied to address healthcare needs of people with disabilities. When inclusion is present in disability policy formulation, care becomes more empathic and congruent with patient values. Good policy requires deep understanding of the issue at hand, and to acquire that understanding we need open, inclusive dialogue with the affected communities - especially vulnerable populations - from formulation through evaluation.

Similarly, surveys on policies impacting vulnerable populations require inclusion. To uncover and report the views and beliefs of members of vulnerable racial groups on race and allocation in a pandemic, surveys ought to be designed and interpreted with inclusion and representation from Black and BAME communities.

The guidance outlined in this review is perhaps best articulated in the form of (1) A Toolkit for Centering Racial Equity Throughout Data Integration (developed at the Actionable Intelligence for Social Policy (AISP) at the University of Pennsylvania) and (2) Cornelius and Harrington's text, A Social

Justice Approach to Survey Design and Analysis. These frameworks outline a series of recommendations to avoid reinforcing “legacies of racist policies and produc[ing] inequitable resource allocation.”

Similarly, a trauma-informed lens seeks to explore what’s “going on with” people instead of what is wrong with them. This view encourages open-ended inquiry and framing of issues through inclusion to more fully understand the nature of a problem. Surveys on hospital policy can certainly integrate trauma-informed principles to more fully tap into patient experience.

The following are a few examples of how the AISP framework - and Cornelius and Harrington’s guidance - could have produced a more informed and generalizable survey.

1. The AISP framework encourages “Clearly discerning who decides how to frame the problem.” Recent descriptions of community preferences for resource allocation in a pandemic by non-Black scholars have failed to accurately “frame the problem” by glossing over the preferences of Black communities. For instance, White and Lo, when citing Biddison’s 2018 study,ⁱⁱⁱ recently claimed:

Most people would agree that patients with an excellent prognosis for survival with treatment should be given priority over those with a poor prognosis on the grounds that, all other things being equal, it is desirable to save more lives rather than fewer.

This is a selective statement. While it is true that most in Biddison’s study did prefer to maximize lives saved, Biddison broke down preferences by race and clearly noted that the majority of her Black participants preferred first-come-first-served (FCFS):

African American participants had significantly lower odds of wanting to always or often use “saving the most life-years” as a criterion for allocation decisions than their white colleagues (OR, 0.34; 95% CI, 0.21-0.58). Conversely, African American participants were significantly more likely to favor often or always using “first come, first served” to drive these key decisions (OR, 2.36; 95% CI, 1.29-4.29).

Given White and Lo’s selective reporting, whenever ethicists frame the views of the public, we should use caution with any extrapolation or summarization especially if the views or experiences of white respondents are dissimilar to – and may drown out - the views of Black or BAME respondents. In this light, I consider how Kappes et al framed their survey questions and the generalizability of their results. Kappes et al report that “participant race did not impact vaccine or ventilator allocation decisions in the US but did impact ventilator allocation attitudes in the UK ($F(4,602) = 6.95, P < 0.001$).” In their conclusion, the authors state, “Our survey identified a broad consensus, in both the

US and UK, for how scarce resources should be allocated during a pandemic.” They also state, “...the vast majority of participants (including those from minority groups) think that in situations of extreme scarcity, life-saving treatment like ventilators should be allocated in a race-neutral way... [a] finding [which] may be relevant to planning for the next pandemic.” These conclusions seem to extrapolate too far.

Response 2.1:

We are grateful to Prof Bruce for encouraging us to re-examine the way in which the framing of the results may have extrapolated too far, and in particular may have diminished the apparent differences in perspectives of Black or BAME respondents.

In the results we had noted that:

“Using mixed linear models [Supplementary Table 1, full results], we find that race of participant impacted ventilator allocation attitudes in the UK ($F(4,602) = 6.95, P < 0.001$) but not the US ($F(6, 635) = 1.38, P = 0.21$). In the UK, Asian and Black participants had more positive attitudes towards race in allocation of ventilators than white participants [SM]. Among BAME participants, 22.1% of participants gave the ventilator to a BAME patient with a 10% lower chance of survival, while only 6.9% of the White participants did.”

In the revised manuscript we have provided further details of the different responses of BAME participants in the UK.

Table 2: Differences in responses between UK White and ethnic minority participants

	Ethnic Minority Participants			White Participants		
	BAME Patient	Coin Toss	White Patient	BAME Patient	Coin Toss	White Patient
Equal Survival Chances	24.4	72.1	3.5	13	73.2	13.8
10% Higher Chance	77.9	11.6	10.5	73.2	15.3	11.5
30% Higher Chance	87.2	10.5	2.3	90.5	2	7.4

10% Lower Chance	22.1	15.1	62.8	6.9	15.5	77.7
30% Lower Chance	4.7	22.1	73.3	1.5	8	90.5

We have revised the conclusion to avoid the risk of over generalising or extrapolating our findings.

Page...

Obviously, the results of this survey do not settle the ethical question of whether governments should prioritise based on race or other forms of social disadvantage. Ethical argument and analysis may lead to prioritisation policies that differ from those endorsed by the majority. However, understanding the general approach of the public towards prioritisation in settings of severe scarcity may be important in debate and deliberation: it may challenge or confirm assumptions about what the public would or would not support, as well as highlight areas of ethical consensus or point to important areas of disagreement.⁴² Our survey identified that in specific allocation scenarios involving high demand and severe resource shortage, a majority of US and UK respondents chose to allocate ventilators to patients with predicted higher chance of survival and vaccines to those at higher risk of severe illness, regardless of patient race. A very small proportion of respondents (<15%) prioritised ventilator allocation to patients from a racial minority, and this was higher among UK respondents from ethnic minority backgrounds. We also found support from a large minority for inclusion of race as a tie-break consideration in vaccine allocation. Although we may hope that it is never needed, these findings may be relevant to planning for the next pandemic.

The survey questions asked whether racial preference ought to be considered under quite specific scenarios. Their questions on ventilator allocation describe scenarios in which two patients need a ventilator and, in that moment, the survey participant needs to choose one patient. These are certainly important questions, but the findings may not more broadly infer that those respondents believe in race neutrality or condone the described allocation system. Any number of reasons may account for participants who may decline to give preferential treatment within Kappes et al's survey but still disagree with the authors' stance of any kind of broad consensus for scarce resource allocation during a pandemic. For instance:

- *Members of vulnerable racial groups may decline to give preferential treatment because racial weighting is controversial and may cause public uproar, while only having limited impact on correcting bias and racism within medicine.*
- *The survey spoke in conclusive terms that may not reflect lived experience – and as such would not be as generalizable. Perhaps, for example, there may be a distinction between responses to the survey's question (patients who have, for example, a 40% chance of*

survival) and a question which could have situated the scenario in a more lifelike and contextual frame (e.g., “a white physician declares a Black or BAME patient has a 40% chance of survival”). Patients (and perhaps especially those from vulnerable populations) often find physicians’ perception of outcomes to be pessimistic, and physician prognosis is imprecise and may be tainted by bias and racism. The survey question, as written, does not reflect the nuance of how these scenarios present themselves, and as such, has limited relevance to future pandemic planning.

Response 2.2:

Thank you for these thoughtful considerations. We have made a number of changes to the manuscript in the light of these important limitations and concerns.

Limitations: We have acknowledged in a new section of the discussion limitations of the study including that the scenarios were artificial, somewhat unrealistic, may not have reflected lived experience, and that responses may have reflected concern about the controversial nature of these questions. See Response E5

Conclusion: We have modified the conclusion to remove inference about wider endorsement of race-neutrality or condoning other aspects of allocation. See Response 2.1

As you note, it is possible that participants would be uneasy about the precise nature of the predictions. This might lead respondents to regard small differences between patients as non-significant.

In fact, we observed that respondents were highly sensitive to small predicted differences in risk. See Response 1.3

Thank you for the thoughtful suggestion that respondents (particularly those with a Black or BAME background) may have been suspicious that doctors’ predictions of outcome for patients from a racial minority would be overly pessimistic and influenced by racial bias. In that case, we would have anticipated that respondents would prioritise patients from a racial minority. Among US respondents (20% of whom were non-white) race did not influence responses to ventilator allocation scenarios. We have expanded discussion of the influence of UK participant race on response to ventilator allocation scenarios. See response 2.1.

We have expanded discussion of other possible explanations of respondents’ reluctance to give additional priority in ventilator allocation to patients from a racial minority. See Response 1.2

“The different weight given to race in the allocation of vaccines versus ventilators in our study may reflect people’s desire to prevent the most deaths.^{7,30} Previous research has shown strong public endorsement for prioritising chances of survival in ventilator allocation and probability of severe illness for vaccine allocation.^{24,27,31–33} However, whereas vaccines are provided to prevent illness, ventilators are provided to those already critically ill. Participants might have felt that allocating ventilators preferentially to those with additional risk factors (race, sex, obesity) could lead to more deaths from COVID-19 if such patients (even with otherwise equal medical risk) have higher mortality rates. Another possible explanation, at least for US participants, is that some were aware of more recent evidence of similar hospitalised critical care outcomes for Black patients.³⁵ This may have led some to conclude that since hospital admissions were in proportion to the occurrence of severe disease in the community, allocation of critical care resources preferentially would not be justified.³⁶ (Since the survey was conducted prior to much of this evidence emerging, this was potentially less likely to influence responses). Respondents might have regarded preferential allocation of a life-saving intervention like ventilators as more legally or ethically questionable than prioritisation of vaccines. Finally, participants might have reasoned that ventilators do not have wider community benefits, but vaccines do. Hence, preferential allocation of vaccines represents a more substantial (and possibly less controversial) means to reduce inequality.”

The AISP framework encourages “Clearly discerning who decides how to... determine what questions to ask.” The survey’s relevance would have been more pronounced if Black and BAME scholars – and community members – participated in crafting the survey questions (and interpreting its results). Perhaps, for instance, Black and BAME community members may have encouraged support for FCFS protocols, as was found in Biddison’s study.

Response 2.3

Thank you for this important suggestion for ways that we could have improved the design of our study. While two of the study authors are themselves from a BAME background, we acknowledge that greater inclusion in the study design may have benefited the study and survey design. We have acknowledged this in the new study limitations section. See Response E.5

Similarly, Cornelius and Harrington recommend a community-inclusive survey development process which may include advance reviews by “Community-based ethics review boards.”^{vi} Perhaps such a review may have mitigated the US sample’s underrepresentation from Black and Hispanic participants which (in Kappes’ et al’s words) may “underestimate how different racial groups feel about allocation of scarce resources.”

Response 2.4

We are grateful for this suggestion. Our survey was reviewed by Social Sciences and Humanities Interdivisional Research Ethics Committee at the University of Oxford, who review a very wide range of research projects and are very sensitive to the importance of inclusion, diversity and sensitivity in research. We have not had access to community-based ethics review in the past, but will explore such an approach for future surveys.

The AISP framework encourages “Including qualitative stories to contextualize quantitative data.” The authors’ survey was limited in terms of the feedback it collected from participants. It did not ask participants why they made their choices. Free-form text is more cumbersome to code and analyze, and perhaps more costly, but it would have provided context and yielded insights on participants’ preferences.

Response 2.5

Thank you for this suggestion. We have added this to the discussion of the study limitations. See Response E.5

In closing, Kappes et al’s paper provides limited contributions to our understanding of public views on race and allocation in a pandemic. The survey design and analysis were not informed by a social justice paradigm, a racial equity lens, or a trauma-informed approach and as such their survey findings have limitations in terms of generalizability and insight. As Cornelius and Harrington state, when we tell “the story of barriers encountered by (vulnerable) populations,” we are behooved to navigate with care.

Response 2.6

We are grateful to Prof Bruce for thoughtfully engaging with our paper. We appreciate that these are sensitive questions, and that others would choose to examine them using alternative methodology and/or to interpret our results in a different way (as does, for example, reviewer 1).

We have sought, in our revision, to highlight the limitations of our study and to avoid over-generalising the insights gleaned from this. Although there is much still to examine and to understand about community attitudes to these controversial questions, we hope that our results may contribute to debate in a positive and constructive way.

VERSION 2 – REVIEW

REVIEWER	Hick , JL University of Minnesota Hennepin-University Partnership, Emergency Medicine
REVIEW RETURNED	24-Jun-2022

GENERAL COMMENTS	Thank you for your revisions to this already strong work. It would be nice to acknowledge somewhere in the paper that a 10% difference in survival had significant meaning for the survey respondents, but that it is impossible clinically to make such fine determinations and thus larger differences in prognosis that would be clinically achievable / relevant should follow this effort. It would also be nice to comment on next steps - figuring out what the drivers behind the allocation decisions were as they are significant, for example. Thank you for this important contribution to our discussion of the ethics and practicalities of considering race and at-risk populations in allocation of scarce resources.
---

REVIEWER	Bruce , Lori Interdisciplinary Center for Bioethics, Yale University, New Haven
REVIEW RETURNED	06-Jul-2022

GENERAL COMMENTS	Summary This document summarizes concerns related to the 2nd draft of the manuscript now titled, "Race and resource allocation: An online survey of US and UK adults' attitudes towards COVID-19 ventilator and vaccine distribution." For the initial review of the first draft, please see Appendix I. In sum, this reviewer thanks the authors for their thoughtful reflections, analysis, and revisions. The authors' revisions (such as the inclusion of Table 2) made notable progress towards meeting the concerns highlighted within the first round of reviews. However, there are still concerns which have not been addressed adequately, and there are additional concerns. First, there are minor grammatical errors throughout. Second, there are concerns related to the content itself, some of which relate to unresolved concerns reported within the first round of comments. I. Grammatical/Syntax Issues Please see separate document "REVISED_MARKED_COPY_LB" for markup.
---

II. Concerns relating to content/argument

Concerns on content and argument are described below and also marked within "REVISED_MARKED_COPY_LB."

1. The references to – and framing of – US racial injustice need revision. For instance:

a. The manuscript refers to the idea of respondents "[sensitivity to] correcting historical injustices" with no mention of current injustices. Current injustices certainly influence respondents' sensitivity to the survey questions.

b. The manuscript reflects on Americans' beliefs without providing the breakdown of beliefs by racial category. Without the breakdown by racial category, the views of Black Americans can be washed away by the views of white people simply because Blacks are not in the majority.

c. As mentioned, the survey was not designed with a social justice framework, and as such, is limited.

Please see comments within "REVISED_MARKED_COPY_LB" to address these (and related) concerns, such as

i. The addition of contextual breakdown for both the Gallup poll and the affirmative action poll;

ii. Reframing of various phrases; and

iii. Citing these two thoughtful resources on social justice approaches to survey design:

1. A Toolkit for Centering Racial Equity Throughout Data Integration (developed at the Actionable Intelligence for Social Policy (AISP) at the University of Pennsylvania); and

2. Cornelius and Harrington's text, A Social Justice Approach to Survey Design and Analysis.

2. To address concerns described within the initial review, but not fully addressed within the 2nd manuscript, add the following additional limitation within the abstract:

"Neither scholars of racial equity nor members of Black/BAME communities were included in the study design or analysis."

3. Manuscript's interpretation of findings

The revised document needs adjustment in various places, primarily for two main reasons which need to be more comprehensively integrated within the abstract and the body of the manuscript.

First, the BAME/Black respondents' differences from white respondents; and secondly, the low participation by Black respondents within the US dataset.

See the attached manuscript for suggested ways of tempering abstract, results, and discussion to reflect what the data articulates. Similarly, please see #3 below.

4. As stated in the initial review, whenever ethicists frame the views of the public, we should use caution with any extrapolation or summarization especially if the views or experiences of white respondents are dissimilar to – and may drown out - the views of Black or BAME respondents.

	The addition of Table 2 is helpful. However, this table highlights distinctions between the white and non-white respondents which are not included within the results section of the abstract. Assuming they are statistically significant, the results section of the abstract needs to be updated to specify the notable differences between BAME/Black and white participants. This includes the 22% of BAME vs. 6.9% of white respondents for the “10% lower chance” scenario – and all other notable differences. The other major differences within Table 2 need to be addressed within the abstract and text, such as “equal survival chances” 24.4 vs 13; and 3.5 vs. 13.8; and notable differences within the “30% lower chance.” 5. Given the differences within Table 2, this data tells an important story that is not being fully expressed within the title, abstract, and body of this manuscript. Consider reframing the manuscript accordingly. 6. It is currently unclear what survey questions are and are not included within Table 2. Table 2 should report both the ventilator responses as well as the vaccine responses. If both sets are not included, please update the table as well as the results & discussion sections to highlight any additional differences. 7. Please also update Table 2 to more fully list the specific survey questions, as the current descriptive text is too brief. 8. Within the current manuscript, the benefit of equality is mentioned without mentioning the benefit of equity. Please see notes within “REVISED_MARKED_COPY_LB.” 9. Re: Response 2.3 Response 2.3 from the authors states: “... ..we acknowledge that greater inclusion in the study design may have benefited the study and survey design. We have acknowledged this in the new study limitations section. See Response E.5.” The acknowledgement, as included within E5, does not fully address the concerns described within the initial review. Lack of more comprehensive involvement of Black/BAME community members should have (1) its own paragraph within the new limitations section. (2) Within the abstract, an additional bullet point needs to be added - add additional limitation on study design following Penn guidance re inclusion from study inception through analysis. Please see “REVISED_MARKED_COPY_LB” for suggested revisions. Appendix I I May 4, 2022 – Initial Manuscript Review
--	--

	This non-blinded review considers Kappes et al's "Public Attitudes Towards Race and Resource Allocation in a Pandemic," submitted to BMJ Open. The author's survey asked members of the public whether vulnerable racial minorities ought to receive preferential weighting for vaccines and ventilators. Given the troubling and extensive racial disparities exhibited during the COVID-19 pandemic, this topic is timely and relevant. This review provides a short list of considerations related to the survey's design and analysis. Survey Design and Analysis Disability advocates often challenge physicians and policymakers with a mantra they adopted from the central European political tradition, "Nothing about us without us." This plea for inclusion speaks to the uninformed and sometimes harmful practices, policies, and laws enacted by the able-bodied to address healthcare needs of people with disabilities. When inclusion is present in disability policy formulation, care becomes more empathic and congruent with patient values. Good policy requires deep understanding of the issue at hand, and to acquire that understanding we need open, inclusive dialogue with the affected communities - especially vulnerable populations - from formulation through evaluation. Similarly, surveys on policies impacting vulnerable populations require inclusion. To uncover and report the views and beliefs of members of vulnerable racial groups on race and allocation in a pandemic, surveys ought to be designed and interpreted with inclusion and representation from Black and BAME communities. The guidance outlined in this review is perhaps best articulated in the form of (1) A Toolkit for Centering Racial Equity Throughout Data Integration (developed at the Actionable Intelligence for Social Policy (AISP) at the University of Pennsylvania) and (2) Cornelius and Harrington's text, A Social Justice Approach to Survey Design and Analysis. These frameworks outline a series of recommendations to avoid reinforcing "legacies of racist policies and produc[ing] inequitable resource allocation." Similarly, a trauma-informed lens seeks to explore what's "going on with" people instead of what is wrong with them . This view encourages open-ended inquiry and framing of issues through inclusion to more fully understand the nature of a problem. Surveys on hospital policy can certainly integrate trauma-informed principles to more fully tap into patient experience. The following are a few examples of how the AISP framework - and Cornelius and Harrington's guidance - could have produced a more informed and generalizable survey.  1. The AISP framework encourages "Clearly discerning who decides how to frame the problem." Recent descriptions of community preferences for resource allocation in a pandemic by non-Black scholars have failed to accurately "frame the problem" by glossing over the preferences of Black communities. For instance, White and Lo, when citing Biddison's 2018 study, recently claimed: Most people would agree that patients with an excellent prognosis for survival with treatment should be given priority over those with a poor prognosis on the grounds that, all other things being equal, it is desirable to save more lives rather than fewer. This is a selective statement. While it is true that most in Biddison's study did prefer to maximize lives saved, Biddison
--	--

broke down preferences by race and clearly noted that the majority of her Black participants preferred first-come-first-served (FCFS):

African American participants had significantly lower odds of wanting to always or often use “saving the most life-years” as a criterion for allocation decisions than their white colleagues (OR, 0.34; 95% CI, 0.21-0.58). Conversely, African American participants were significantly more likely to favor often or always using “first come, first served” to drive these key decisions (OR, 2.36; 95% CI, 1.29-4.29).

Given White and Lo’s selective reporting, whenever ethicists frame the views of the public, we should use caution with any extrapolation or summarization especially if the views or experiences of white respondents are dissimilar to – and may drown out - the views of Black or BAME respondents.

In this light, I consider how Kappes et al framed their survey questions and the generalizability of their results. Kappes et al report that “participant race did not impact vaccine or ventilator allocation decisions in the US but did impact ventilator allocation attitudes in the UK ($F(4,602) = 6.95, P < 0.001$).” In their conclusion, the authors state, “Our survey identified a broad consensus, in both the US and UK, for how scarce resources should be allocated during a pandemic.” They also state, “...the vast majority of participants (including those from minority groups) think that in situations of extreme scarcity, life-saving treatment like ventilators should be allocated in a race-neutral way... [a] finding [which] may be relevant to planning for the next pandemic.” These conclusions seem to extrapolate too far.

The survey questions asked whether racial preference ought to be considered under quite specific scenarios. Their questions on ventilator allocation describe scenarios in which two patients need a ventilator and, in that moment, the survey participant needs to choose one patient. These are certainly important questions, but the findings may not more broadly infer that those respondents believe in race neutrality or condone the described allocation system.

Any number of reasons may account for participants who may decline to give preferential treatment within Kappes et al’s survey but still disagree with the authors’ stance of any kind of broad consensus for scarce resource allocation during a pandemic. For instance:

- Members of vulnerable racial groups may decline to give preferential treatment because racial weighting is controversial and may cause public uproar, while only having limited impact on correcting bias and racism within medicine.
- The survey spoke in conclusive terms that may not reflect lived experience – and as such would not be as generalizable. Perhaps, for example, there may be a distinction between responses to the survey’s question (patients who have, for example, a 40% chance of survival) and a question which could have situated the scenario in a more lifelike and contextual frame (e.g., “a white physician declares a Black or BAME patient has a 40% chance of survival”). Patients (and perhaps especially those from vulnerable populations) often find physicians’ perception of outcomes to be pessimistic, and physician prognosis is imprecise

and may be tainted by bias and racism. The survey question, as written, does not reflect the nuance of how these scenarios present themselves, and as such, has limited relevance to future pandemic planning.

2. The AISP framework encourages “Clearly discerning who decides how to... determine what questions to ask.” The survey’s relevance would have been more pronounced if Black and BAME scholars – and community members – participated in crafting the survey questions (and interpreting its results). Perhaps, for instance, Black and BAME community members may have encouraged support for FCFS protocols, as was found in Biddison’s study.

Similarly, Cornelius and Harrington recommend a community-inclusive survey development process which may include advance reviews by “Community-based ethics review boards.” Perhaps such a review may have mitigated the US sample’s underrepresentation from Black and Hispanic participants which (in Kappes’ et al’s words) may “underestimate how different racial groups feel about allocation of scarce resources.”

3. The AISP framework encourages “Including qualitative stories to contextualize quantitative data.” The authors’ survey was limited in terms of the feedback it collected from participants. It did not ask participants why they made their choices. Free-form text is more cumbersome to code and analyze, and perhaps more costly, but it would have provided context and yielded insights on participants’ preferences.

In closing, Kappes et al’s paper provides limited contributions to our understanding of public views on race and allocation in a pandemic. The survey design and analysis were not informed by a social justice paradigm, a racial equity lens, or a trauma-informed approach and as such their survey findings have limitations in terms of generalizability and insight. As Cornelius and Harrington state, when we tell “the story of barriers encountered by (vulnerable) populations,” we are behooved to navigate with care.

Actionable Intelligence for Social Policy. “A Toolkit for Centering Racial Equity Throughout Data Integration.” University of Pennsylvania. <https://aisp.upenn.edu/centering-equity/>. Last accessed May 3, 2022.

Lanphier, Elizabeth, and Uchenna E. Anani. "Trauma informed ethics consultation." *The American Journal of Bioethics* (2021): 1-13.

Biddison ELD, Gwon HS, Schoch-Spana M, et al. Scarce resource allocation during disasters: a mixed-method community engagement study. *Chest*. 2018;153(1):187-195. doi:10.1016/j.chest.2017.08.001

White, Douglas B., and Bernard Lo. "Mitigating inequities and saving lives with ICU triage during the COVID-19 pandemic." *American Journal of Respiratory and Critical Care Medicine* 203, no. 3 (2021): 287-295.

Biddison et al, 192.

Cornelius, Llewellyn Joseph, and Donna Harrington. A social justice approach to survey design and analysis. *Pocket Guide to Social Work Re*, 2014. See p. 118.

VERSION 2 – AUTHOR RESPONSE

Reviewer: 1

Dr. JL Hick, University of Minnesota Hennepin-University Partnership

Comments to the Author:

Thank you for your revisions to this already strong work. It would be nice to acknowledge somewhere in the paper that a 10% difference in survival had significant meaning for the survey respondents, but that it is impossible clinically to make such fine determinations and thus larger differences in prognosis that would be clinically achievable/relevant should follow this effort.

Response: Thank you for this suggestion. We now write in the discussion (added text highlighted, page 8):

“They were highly sensitive to small differences in predicted outcome. For example, for a 10% difference in predicted survival, 71.9% of participants gave the ventilator to the patient with higher chances while 82.7% chose to toss a coin where there was no difference in predicted chance. For vaccine allocation we found similar results for even smaller differences in risk of severe Covid (5% difference). (It is important to note that such small differences in prognosis are clinically impossible to achieve, but demonstrate participants' sensitivity to medical factors.)

It would also be nice to comment on next steps - figuring out what the drivers behind the allocation decisions were as they are significant, for example.

Response: Thank you for this suggestion. We now write in the discussion (page 10):

One promising future direction would be to use computational models of decision-making to quantify the weight people give in their allocation decision to different medical and social factors when presented simultaneously, as is the case in clinical practice.

Thank you for this important contribution to our discussion of the ethics and practicalities of considering race and at-risk populations in allocation of scarce resources.

Reviewer: 2

Prof. Lori Bruce, Interdisciplinary Center for Bioethics, Yale University, New Haven

2nd Review of bmjopen-2022-062561.R1 Summary

This document summarizes concerns related to the 2nd draft of the manuscript now titled, "Race and resource allocation: An online survey of US and UK adults' attitudes towards COVID-19 ventilator and vaccine distribution." For the initial review of the first draft, please see Appendix I.

In sum, this reviewer thanks the authors for their thoughtful reflections, analysis, and revisions. The authors' revisions (such as the inclusion of Table 2) made notable progress towards meeting the concerns highlighted within the first round of reviews. However, there are still concerns which have not been addressed adequately, and there are additional concerns.

First, there are minor grammatical errors throughout.

Second, there are concerns related to the content itself, some of which relate to unresolved concerns reported within the first round of comments.

I. Grammatical/Syntax Issues

Please see separate document "REVISED_MARKED_COPY_LB" for markup.

Response: Thanks for the careful reading. We now changed "could" to "may possibly" on page 3, deleted "is" on page 5, capitalised "Modern Racism Scale" (various places), and added a percentage sign (page 6).

II. Concerns relating to content/argument

Concerns on content and argument are described below and also marked within "REVISED_MARKED_COPY_LB."

1. The references to – and framing of – US racial injustice need revision. For instance:

a. The manuscript refers to the idea of respondents "[sensitivity to] correcting historical injustices" with no mention of current injustices. Current injustices certainly influence respondents' sensitivity to the survey questions.

Response: We deleted the word historical to correct this (page 10).

b. The manuscript reflects on Americans' beliefs without providing the breakdown of beliefs by racial category. Without the breakdown by racial category, the views of Black Americans can be washed away by the views of white people simply because Blacks are not in the majority.

Response: Thank you for this suggestion. We have reviewed the differences in responses by racial group and have provided some additional details in the manuscript as discussed below. It is important to note that we did not find statistical differences between White Americans and Black Americans. This is potentially because of the small number of Black Americans in our sample, though the sample size was powered to find small differences (see response 4 below). Because the differences are not significant, the suggested breakdown by race is potentially misleading. As a consequence we would ordinarily not report the breakdown, but in recognition that this may be of interest to other readers have included them in the supplemental materials.

c. As mentioned, the survey was not designed with a social justice framework, and as such, is limited.

Please see comments within "REVISED_MARKED_COPY_LB" to address these (and related) concerns, such as

i. The addition of contextual breakdown for both the Gallup poll and the affirmative action poll

Response: We have deleted the section on the Gallup poll (page 11).

ii. Reframing of various phrases;

and

iii. Citing these two thoughtful resources on social justice approaches to survey

design:

1. A Toolkit for Centering Racial Equity Throughout Data Integration

(developed at the Actionable Intelligence for Social Policy (AISP) at the University of Pennsylvania); and

2. Cornelius and Harrington's text, A Social Justice Approach to Survey

Response: We now cite the two frameworks in the limitations section of the manuscript, writing:

Scenarios were highly specific, somewhat artificial, and clinically unrealistic (particularly in relation to small predicted differences between patients or groups). This could have differed from participants'

lived experience and may have contributed to unease in prioritisation decisions. The survey design did not follow an explicit social justice framework^{29,30}”.

Design and Analysis.

2. To address concerns described within the initial review, but not fully addressed within the 2nd manuscript, add the following additional limitation within the abstract:

“Neither scholars of racial equity nor members of Black/BAME communities were included in the study design or analysis.”

Response: To highlight that non-academic members of the Black and BAME community were not involved, and to avoid any confusion stemming from the term BAME, we now include the following sentence “Members of non-academic minoritised groups were not included in the study design or analysis.” as a key limitation of the study.

3. Manuscript’s interpretation of findings

The revised document needs adjustment in various places, primarily for two main reasons which need to be more comprehensively integrated within the abstract and the body of the manuscript.

First, the BAME/Black respondents’ differences from white respondents; and secondly, the low participation by Black respondents within the US dataset.

See the attached manuscript for suggested ways of tempering abstract, results, and discussion to reflect what the data articulates. Similarly, please see #3 below.

Response: To address BL1 in REVISED_MARKED_COPY_LB, we included the number of Black American participants (114 participants) and BAME participants (139 participants). Note that BAME stands for Black and Asian Ethnic Minority, includes Asian minorities (the largest ethnic minority in the UK), and does not refer to only Black British people (as your comment suggests). This might also help to explain comment BL18 in REVISED_MARKED_COPY_LB. Reviewer 2 mentions here that the percentage of BAME respondents appears to be “quite small”, and whether this national representative. Accordingly to the last census in the UK, 3.2% of the UK population belong to a Black ethnic group (3.2% in our sample) and 7.5% were from Asian ethnic groups (7.6% in our sample).

4. As stated in the initial review, whenever ethicists frame the views of the public, we should use caution with any extrapolation or summarization especially if the views or experiences of white respondents are dissimilar to – and may drown out - the views of Black or BAME respondents.

The addition of Table 2 is helpful. However, this table highlights distinctions between the white and non-white respondents which are not included within the results section of the abstract. Assuming they are statistically significant, the results section of the abstract needs to be updated to specify the notable differences between BAME/Black and white participants. This includes the 22% of BAME vs. 6.9% of white respondents for the “10% lower chance” scenario – and all other notable differences.

The other major differences within Table 2 need to be addressed within the abstract and text, such as “equal survival chances” 24.4 vs 13; and 3.5 vs. 13.8; and notable differences within the “30% lower chance.”

Response: To address this comment and BL2 in REVISED_MARKED_COPY_LB, we now include the UK BAME percentages for the ventilator decisions (where we find a statistical difference). At the start of the results section of the abstract we now write:

“Participant race did not impact vaccine or ventilator allocation decisions in the US, but did impact ventilator allocation attitudes in the UK ($F(4,602) = 6.95, P < 0.001$). When asked to allocate ventilators to a racial minority or White patient with identical chances of survival, 14.8% allocated to the minority patient (UK BAME participants: 24.4%), 16.2% chose the White patient, and 68.9% chose to toss a coin. When the racial minority patient had a 10% lower chance of survival, 12.4% participants allocated the ventilator to the minority patient (UK BAME participants: 22.1%).”

To address BL3 in REVISED_MARKED_COPY_LB, we now start the results section with the results concerning participants’ race (see above). To address the second part of the comment (“but since Black participants within the US sample were underrepresented, this finding may be inconclusive.”), we performed a sensitivity analysis which shows that our sample gives us .99 statistical power (99% chance to find an effect if there is an effect) to detect even a small difference between White and Black American participants. We now added to the following sentence to the result section:

A sensitivity analysis showed that the American sample afforded us a power of .99 to find a small difference between White and Black participants.

5. Given the differences within Table 2, this data tells an important story that is not being fully expressed within the title, abstract, and body of this manuscript. Consider reframing the manuscript accordingly.

Response: Thank you for this suggestion. We have added in details of this difference in response in both the abstract and the body of the text. We have added the following to the start off the discussion.

“Relatively few respondents in our survey, even those from racial minorities themselves, endorsed priority for ventilators for those from racial minorities. However, BAME respondents in the UK (but not Black respondents in the US) were more inclined than White respondents to preferentially allocate a ventilator to a patient from a racial minority.”

Our conclusion includes reference to this finding:

“Our survey identified that in specific allocation scenarios involving high demand and severe resource shortage, a majority of US and UK respondents chose to allocate ventilators to patients with predicted higher chance of survival and vaccines to those at higher risk of severe illness, regardless of patient race. A very small proportion of respondents (<15%) prioritised ventilator allocation to patients from a racial minority, though this was higher among UK respondents from ethnic minority backgrounds.”

6. It is currently unclear what survey questions are and are not included within Table 2. Table 2 should report both the ventilator responses as well as the vaccine responses. If both sets are not included, please update the table as well as the results & discussion sections to highlight any additional differences.

Response: We report all scenarios involving ventilators as well as race for UK participants since these are the only scenarios that show a significant effect of participants' race.

As noted in response 1 above, we have added in additional details in the supplemental materials.

7. Please also update Table 2 to more fully list the specific survey questions, as the current descriptive text is too brief.

Response: We now write: “Table 2. Differences in responses between UK White and BAME participants for all scenarios involving ventilators and race”

8. Within the current manuscript, the benefit of equality is mentioned without mentioning the benefit of equity. Please see notes within “REVISED_MARKED_COPY_LB.”

Response:

Thank you for highlighting this. We have added this to the introduction:

“Furthermore, in some circumstances, preferential allocation to members of a disadvantaged group may possibly increase overall mortality from COVID-19^{22,23}. On the other hand, not considering it could widen the race-based difference in COVID-19 deaths and conflict with the principle of equity.”

9. Re: Response 2.3

Response 2.3 from the authors states:

“... ..we acknowledge that greater inclusion in the study design may have benefited the study and survey design. We have acknowledged this in the new study limitations section. See Response E.5.”

The acknowledgement, as included within E5, does not fully address the concerns described within the initial review.

Lack of more comprehensive involvement of Black/BAME community members should have (1) its own paragraph within the new limitations section.

Response: We agree. We now added the following paragraph in our discussion section:

Another limitation is the lack of involvement of non-academic members of the Black and BAME community. Recent frameworks of survey development encourage active involvement of racial minorities in the development and analysis of research as best practice to ensure, for instance, that the lived experiences of community members are represented when designing the survey and that different perspectives of how the same data could be interpreted are considered. Hence, it is important to acknowledge that our questions, data collection, and interpretation only represent one perspective and that involvement of non-academic Black and BAME community members might have produced different results.

(2) Within the abstract, an additional bullet point needs to be added - add additional limitation on study design following Penn guidance re inclusion from study inception through analysis.

Please see “REVISED_MARKED_COPY_LB” for suggested revisions.

Response: See response 2 above.

VERSION 3 – REVIEW

REVIEWER	Bruce , Lori Interdisciplinary Center for Bioethics, Yale University, New Haven
-----------------	--

REVIEW RETURNED	14-Oct-2022
GENERAL COMMENTS	These final revisions reflect the study's unique contributions as well as study design limitations.